# An umpolung strategy to react catalytic enols with nucleophiles

Amparo Sanz-Marco [1], Samuel Martinez-Erro [1], Martin Pauze [1,2], Enrique Gómez-Bengoa [2] & Belén Martín-Matute [1]*

The selective synthesis of α-functionalized ketones with two similar enolizable positions can be accomplished using allylic alcohols and iridium(III) catalysts. A formal 1,3-hydrogen shift on allylic alcohols generates catalytic iridium-enolates in a stereospecific manner, which are able to react with electrophiles to yield α-functionalized ketones as single constitutional isomers. However, the employment of nucleophiles to react with the nucleophilic catalytic enolates in this chemistry is still unknown. Herein, we report an umpolung strategy for the selective synthesis of α-alkoxy carbonyl compounds by the reaction of iridium enolates and alcohols promoted by an iodine(III) reagent. Moreover, the protocol also works in an intramolecular fashion to synthesize 3(2H)-furanones from γ-keto allylic alcohols. Experimental and computational investigations have been carried out, and mechanisms are proposed for both the inter- and intramolecular reactions, explaining the key role of the iodine(III) reagent in this umpolung approach.

[1] Department of Organic Chemistry, Stockholm University, Stockholm SE-10691, Sweden. [2] Departamento de Química Orgánica I, Universidad del País Vasco/UPV-EHU, Manuel de Lardizabal 3, Donostia – San Sebastián 20018, Spain. *email: belen.martin.matute@su.se

Umpolung reactions represent a powerful approach for the introduction of functional groups into organic molecules where this would not otherwise be possible due to electronic mismatch[1]. This term was introduced by D. Seebach and E. J. Corey to refer the inversion of reactivity of acyl carbon atoms in their reactions with electrophiles[2]. Analogous methods based on switching the reactivity of amines[3], imines[4–6], or carbonyl groups[7–13] have been developed in recent decades. This approach has played a pivotal role in the design of synthetic procedures, allowing access to target molecules that would be difficult to obtain by classical processes.

One example of a reaction where it is necessary to use an umpolung approach is the functionalization of ketones in the α position using nucleophiles[14]. To achieve this goal, iodine(III) compounds[15–18], transition metals[19], Lewis acids[20,21], and other reagents[22] have been used. For the synthesis of α-functionalized ketones with iodine(III) compounds, several reagents have been successfully used. These include Koser's reagent[23], (diacetoxyiodo)benzene (PIDA)[24], p-iodotoluene difluoride[25], and benzioxol(on)es (BX)[26]. When acyclic I(III) reagents are used, the nucleophile can be either an external nucleophile or a ligand of the I(III) center. However, for cyclic hypervalent iodine reagents, only examples where the nucleophile is part of the reagent have been reported, what requires an enormous synthetic effort as every reaction requires the synthesis of a new I(III) reagent[27–30]. In terms of the reaction substrates, dicarbonyl compounds, aromatic and cyclic ketones, and silyl enol ethers have all been used. However, no regiocontrol was achieved for ketones containing two enolizable α carbons with similar electronic/steric properties (Fig. 1a)[31].

Allylic alcohols have been proven to be a very useful class of compounds as ketone synthons during the last decades[32,33]. Our group has reported the use of allylic alcohols as enolate equivalents for the preparation of α-functionalized carbonyl compounds as single constitutional isomers[34–41]. This approach relies on the generation of iridium enolates as catalytic intermediates through 1,3-hydrogen transfer, and the in situ reaction of these intermediates with a variety of heteroatomic electrophiles, including halogen and oxygen-based electrophilic species. The stoichiometric synthesis of enolates is avoided, and the reaction takes place under base-free conditions; as a result, a single substituent may be introduced at the desired α carbon. This is difficult to achieve starting from ketones with no clear electronic or steric bias (Fig. 1b). This approach has been successful, but it relies on the use of heteroatomic electrophiles. These are highly reactive species, and are less readily available than their nucleophilic counterparts. In this paper, we report an approach that inverts the polarity of the iridium enolate intermediate, allowing it to react with O-nucleophiles (Fig. 1c). In a reaction mediated by an iridium(III) complex, allylic alcohols react with nucleophiles to yield α-alkoxy ketones as single constitutional isomers. The reaction is mediated by 1-fluoro-3,3-dimethyl-1,3-dihydro-1λ³-benzo[d][1,2]iodaoxole. Furthermore, under the same conditions β-keto allylic alcohols cyclize yielding 3(2H)-furanones.

## Results

**Reaction development and optimization.** We focused on the use of alcohols as nucleophiles; this would lead to the formation of α-alkoxy ketones. α-Alkoxy ketones are important building blocks for synthesis, and they are also present in many natural products[42,43]. We selected allylic alcohol **1a** as a model substrate and methanol as a nucleophile. A preliminary screening of iodine(III) reagents (Fig. 2) revealed that 1-fluoro-3,3-dimethyl-1,3-dihydro-1λ³-benzo[d][1,2]iodaoxole (**I**) gave α-methoxy ketone **2a** in a promising 43% yield as a single constitutional isomer (see Supplementary Table 1). In contrast, **II–IX** resulted in lower yields, ranging from 0 to 15%. **I** is a stable compound, and it is commercially available and easy to handle. Taking all this into account, **I** was chosen for further optimization studies in combination with the commercially available [Cp*IrCl₂]₂[44,45].

It should be noted that the selective formation of the desired product **2** in this reaction represents an enormous challenge (Table 1). Several by-products may be formed from the allylic alcohol substrates **1**, including unsubstituted ketone **3**, and also enone **4**. Furthermore, other nucleophiles such as water could

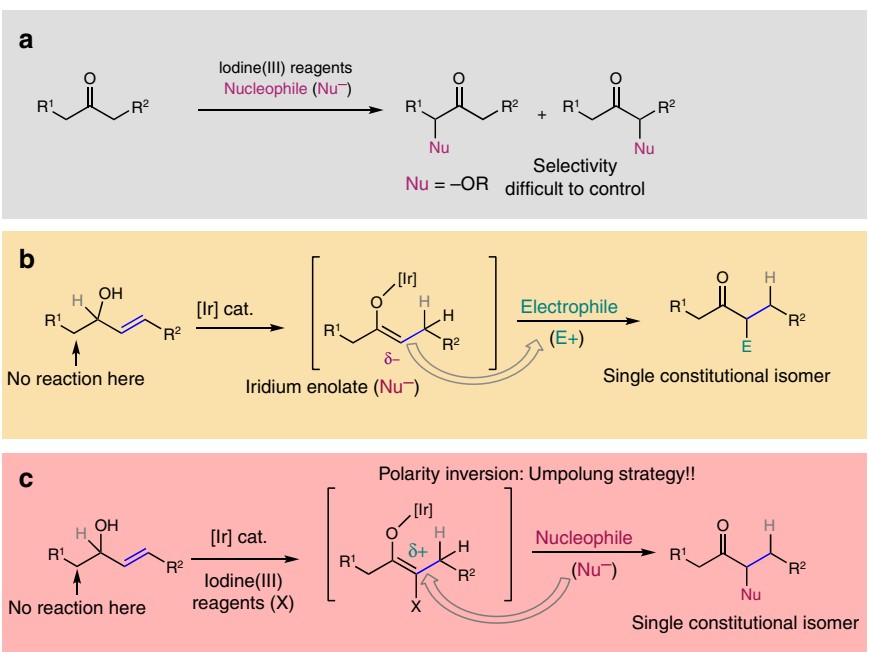

**Fig. 1** Synthesis of α-substituted carbonyl compounds. **a** α-Functionalization of ketones through umpolung reactions. **b** Isomerization/functionalization of allylic alcohols with electrophiles. **c** Our approach: Isomerization/umpolung functionalization of allylic alcohols

**Fig. 2** Iodine(III) reagents screened. Yields of **2a** determined by [1]H NMR spectroscopy using an internal standard (1,2,4,5-tetrachloro-3-nitrobenzene) (Supplementary Table 1)

also react, leading to the formation of α-hydroxy ketone **5**. All these by-products were observed in our initial test reactions (Table 1, entry 1 and Supplementary Table 1), where common solvents previously used in related reactions were tested[35–41]. When either acetone or THF was used (Table 1, entries 1 and 2), **2a** was formed in moderate yield (up to 43%) together with by-products **3–6** in yields ranging from 6 to 12%. Further testing led us to the use of 2,2,2-trifluoroethanol (TFE) (entries 3–5), which gave α-methoxy ketone **2a** in 54% yield along with the same by-products in a similar ratio (Table 1, entry 4). A slightly better yield of **2a** was obtained when the temperature was raised to 35 °C (57%; Table 1, entry 6). Increasing the temperature further did not prove beneficial (Table 1, entry 7). Remarkably, **2a** was formed in 75% yield when the reaction mixture was diluted (from 0.2 to 0.02 M). The amounts of by-products **3** and **4** decreased, and the formation of **5** was suppressed (Table 1, entry 8). When the amount of KBF4 was increased (from 0.3 to 0.8 equiv.), **2a** was formed in an excellent yield of 89% (Table 1, entry 9). The use of other additives failed to give higher yields of **2a** (Table 1, entries 10 and 11). No conversion was observed when the chloride-free [Cp*Ir(H$_2$O)$_3$]SO$_4$ complex was used as the catalyst. It has previously been shown that a halide ligand is essential in the isomerization of allylic alcohols[45]. In fact, when a chlorinated agent such as N-chlorosuccinimide is added in combination with [Cp*Ir(H$_2$O)$_3$]SO$_4$, the desired product is observed (Supplementary Fig. 4). Control experiments were also carried out (Table 1, entries 13–15). In the absence of the iridium catalyst, allylic alcohol **1a** was recovered in 86% yield (Table 1, entry 13), and **2a** was not detected. The presence of the additive as well as TFE as cosolvent was necessary for the desired product **2a** to be formed in high yield (Table 1, entry 9 vs entries 14 and 15). Saturated ketone **3** was obtained in quantitative yield when the reaction was carried out in the absence of **I** (Table 1, entry 16). Thus, we found that the umpolung reaction of **1a** was best carried out using **I** in a TFE/MeOH mixture with KBF4 as additive at 35 °C catalyzed by [Cp*IrCl$_2$]$_2$ at a 0.02 M concentration of the allylic alcohol (Table 1, entry 9). We went on to study the substrate scope of the reaction under these optimal reaction conditions.

**Reaction scope.** For our studies of the substrate scope of the reaction (Fig. 3), we focused on allylic alcohols that would lead to

α-methoxy ketones that are not accessible by the alternative direct α-functionalization of ketones, due to poor regioselectively. Allylic alcohols bearing terminal double bonds generally gave the corresponding α-methoxy ketones **2a–2n** in moderate to good yields. Steric effects play an important role, and a more hindered allylic alcohol **1d** gave **2d** in only 45% yield. Aromatic allylic alcohols were found to be well tolerated, and **1e** gave a 64% yield of **2e**. Substrates bearing other functional groups such as an alkene, an ether, or a silyl ether (**1f–1h**) gave the corresponding products in excellent yields (84%, 99%, and 80%, respectively). This highlights the functional-group compatibility of the method. Remarkably, the reaction is chemoselective for the allylic alcohol functionality. Other functional groups with acidic α-methylene groups remain untouched. Allylic alcohol substrates bearing additional ketone (**2i**, 42% yield), nitrile (**2j**, 91% yield), or sulfone groups (**2k**, 74% yield) all gave the desired products with the reaction only taking place at the allylic alcohol. Furthermore, the reaction of chloride-containing allylic alcohol **1l** proceeded in high yield (89%), and also diverse functionalized α-methoxy ketones, such as azide **2m** (69%) and morpholine **2n** (77%), were prepared in good yields from **1l** in a one-pot two-step procedure. By simply selecting the starting allylic alcohol, constitutional isomers were prepared selectively. For instance, α-methoxy ketones **2b** and **2q** were selectively synthesized from allylic alcohols **1b** (external double bond) and **1q** (internal double bond), respectively. Allylic alcohols with internal 1,2-dis-ubstituted double bonds (**1o–1u**) also reacted smoothly to give generally good yields of the corresponding products. Other alcohols such as ethanol and propanol afforded also the corresponding products, α-ethoxy ketone **6a** in 40% and α-propoxy ketone **7a** in 20% yield, respectively. In all examples shown in Fig. 3, the products were obtained as single constitutional isomers. Remarkably, this efficient umpolung protocol was extended to primary allylic alcohols (**1v–1w**) to give α-methoxy aldehydes in high yields. The method was also extended to more complex molecules derived from trans-androsterone (**1x**) and lythocolic acid (**1y**), which bear multiple functional groups such as esters and ketones, in addition to several stereocenters, and the desired products were obtained in high yields.

We also found that the reaction could be carried out in an intramolecular manner, with the oxygen of a carbonyl group acting as the nucleophile (Fig. 4). Thus, allylic alcohols **8a–8h**,

**Table 1 Optimization studies[a]**

| Entry | Solvent | T [°C] | Additive | Yield [%][b] 2a/3/4/5 |
|---|---|---|---|---|
| 1 | Acetone | 23 | KBF$_4$ | 43/12/12/10 |
| 2 | THF | 23 | KBF$_4$ | 27/11/6/12 |
| 3 | HFIP[c] | 23 | KBF$_4$ | 45/-/29/- |
| 4 | TFE | 23 | KBF$_4$ | 54/9/10/10 |
| 5 | PhCF$_3$ | 23 | KBF$_4$ | 52/6/7/9 |
| 6 | TFE | 35 | KBF$_4$ | 57/10/10 |
| 7 | TFE | 45 | KBF$_4$ | 48/7/2/- |
| 8[d] | TFE | 35 | KBF$_4$ | 75/8/2/- |
| 9[d,e] | TFE | 35 | KBF$_4$ | 89/4/6/- |
| 10[d,e] | TFE | 35 | NaBF$_4$ | 85/7/5/- |
| 11[d,e] | TFE | 35 | TBAF | 65/20/-/10 |
| 12[d,e,f] | TFE | 35 | KBF$_4$ | - |
| 13[d,e,g] | TFE | 35 | KBF$_4$ | - |
| 14[d] | TFE | 35 | - | 67/5/3/10 |
| 15[d,e] | - | 35 | KBF$_4$ | 68/5/2/11 |
| 16[d,e,h] | TFE | 35 | KBF$_4$ | -/99/-/- |

[a]Unless otherwise noted, all experiments were carried out under an atmosphere of air on a scale of 0.15 mmol of **1a** (0.2 M), with KBF$_4$ (0.3 equiv.) for 2 h
[b]Determined by $^1$H NMR spectroscopy using an internal standard (1,2,4,5-tetrachloro-3-nitrobenzene)
[c]HFIP = 1,1,1,3,3,3-hexafluoroisopropanol
[d]0.02 M instead of 0.2 M
[e]0.8 equiv. of additive instead of 0.3 equiv
[f][Cp*Ir(H$_2$O)$_3$]SO$_4$ instead of [Cp*IrCl$_2$]$_2$
[g]In the absence of catalyst, 86% of **1a** was recovered
[h]In the absence of **I**

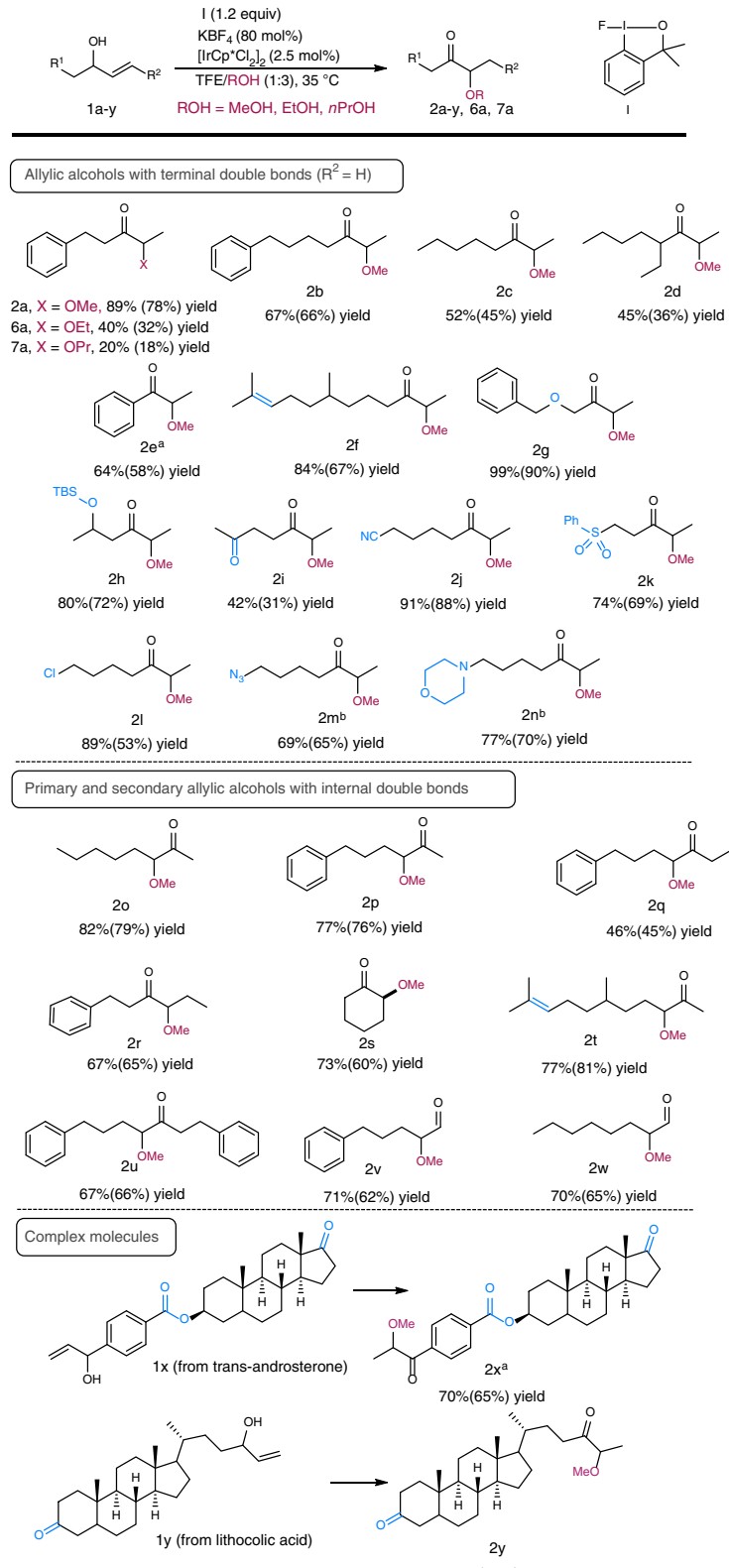

**Fig. 3** Scope of allylic alcohols **1**. Yields by [1]H NMR spectroscopy (isolated yields in parentheses). **a** By slow addition of the reactants. **b** From **2l** in a one-pot two-step procedure

which have a ketone group in a 1,3-relationship with the alcohol functionality, were efficiently transformed into 3(2H)-furanones **9**. This five-membered ring system is an important structural unit, and it can be found in a large number of natural

products and other compounds with applications in medicine and biology[46–49].

Allylic alcohol **8a**, with R³ = H, gave 3(2H)-furanone **9a** in 65% yield. The introduction of a substituent at R³ had a

**Fig. 4** Scope of allylic alcohols **8**. Yields by [1]H NMR spectroscopy (isolated yields in parentheses)

significant positive effect on the yield of the tandem reaction. Thus, alcohols **8b**–**8f** gave the corresponding 3(2H)-furanones in excellent yields. Furthermore, allylic alcohols containing a cyclic ketone moiety **8g** and **8h** reacted well to give bicyclic products **9g** and **9h** (as a mixture of two diastereomers). The intramolecular umpolung reaction was extended to allylic alcohols **8i**–**8k**, which contain an amide group that can act as an internal nucleophile. These substrates gave highly functionalized aminofuranones in excellent yields. Our umpolung strategy allows the construction of these highly important heterocycles from easily accessible starting materials.

**Mechanism**. To gain some insight into the reaction mechanism, and in particular into the role of the iodine(III) reagent **I**, we carried out DFT calculations at the M06/6-31G(d,p) level using the Gaussian 16 suit of programs (see the Supplementary Figs. 7 and 8, and Supplementary Data 1). In analogy with our previous work on the [Cp*Ir(III)]-catalyzed isomerization of allylic alcohols[34,37,45,50] we propose that the reaction starts with the allylic alcohol undergoing a hydrogen-transfer step (1,3-hydride shift) mediated by the metal catalyst. This leads to iridium enolate species **A** (represented as an $\eta^3$-enolate in Fig. 5)[51]. Our hypothesis is that the role of the additive (KBF$_4$) is to increase the rate of the formation of iridium enolate species **A**. We carried out kinetic studies on the isomerisation of allylic alcohols, and found that the reaction is faster in the presence of KBF$_4$ than in its absence (Supplementary Fig. 1). The initial reaction rate of **1a**-d was found to be comparable to that of **1a**, indicating that the 1,3-H shift is not rate determing (Supplementary Fig. 5).

The reaction of enolate **A** with **I** forms an enolonium intermediate **B**, which then reacts further with MeOH to form enolonium **C**[52,53]. From **C**, the final product **2** is obtained by reductive ligand coupling via **TS1**[28,54]. This step has an activation energy of 16.2 kcal mol$^{-1}$, which is perfectly attainable under the experimental reaction conditions. In the computational model of **TS1**, a molecule of TFE was introduced to activate the carbonyl group of the substrate by hydrogen bonding, and thus lower the activation barrier. When this step was modeled in the absence of TFE, the activation energy increased to 21.8 kcal mol$^{-1}$ (Supplementary Fig. 8). Enolonium intermediates **B**′ and **C**′, tautomers containing an I–O bond instead of an I–C bond were also considered, but these species were excluded as they have higher

energies than **B** and **C** ($\Delta G = 14.1$ and 5.3 kcal mol$^{-1}$, respectively). The energies of other isomeric forms of the enolonium intermediate (with a different arrangement of the substituents around the I(III) center) were also calculated, and these were all found to be much higher in energy (Supplementary Fig. 7). Besides, we could not locate any transition structure to form **2** starting from tautomer **C**′. Thus, **C** was the most plausible intermediate for this mechanism.

To investigate whether the iridium complex or KBF$_4$ are involved in the reaction mechanism after enolonium **B** is formed, we performed control experiments in the laboratory where a pre-formed silyl enol ether was used as the starting material instead of the allylic alcohol. Neither the presence of KBF$_4$ nor that of the iridium catalyst in the reactions from the silyl enol ether had any effect on the yield of the product (Supplementary Figs. 2 and 3). This suggests that none of them participates in the mechanism after enolonium **B** has been formed.

Further, reactions tested in the presence of radical scavengers, such as 2,2,6,6-tetramethylpiperidin-1-yl)oxidanyl (TEMPO) or 2-diphenylethylene, afforded the product (**2a**) in high yields (Supplementary Fig. 6). This suggests that a single-electron transfer mechanism is not operating.

We also investigated the mechanism of the cyclization reaction of type-**8** allylic alcohols. An analogous enolonium species **D** is transformed into enolonium **E** by a downhill nucleophilic-addition/proton-transfer process ($\Delta G = -11.7$ kcal mol$^{-1}$, Fig. 6). The intramolecular displacement of the aryliodonium group via **TS2** leads to the final (2H)-furan-3-one **9** in a very fast step, with a calculated activation energy of 8.0 kcal mol$^{-1}$. Such intramolecular nucleophilic displacement reactions of aryliodonium groups have previously been suggested by Jacobsen and coworkers among others[55,56].

We also investigated why we never observed α-methoxylated products when we used **8**-type substrates in our experimental work, and found out that starting from **D**, the competing transition state leading to the α-methoxy carbonyl compound (**TS3**) is higher in energy than **TS2** by ca. 10 kcal mol$^{-1}$ (18.2 kcal mol$^{-1}$, Fig. 6).

**Discussion**

We have described a selective umpolung strategy which involves the reaction of an enolate species, formed from an allylic alcohol

**Fig. 5** Proposed intermolecular reaction mechanism. DFT calculations for the umpolung reaction of allylic alcohols with methanol. Values correspond to Gibbs free energies in kcal mol$^{-1}$

**Fig. 6** Proposed intramolecular reaction mechanism. **a** DFT calculations for the umpolung reaction of γ-keto allylic alcohols. **b** TS for the α-methoxylation of γ-keto allylic alcohols

under catalytic conditions, with methanol. The catalytic enolate intermediate is formed by a 1,3-hydride shift mediated by an iridium catalyst. The two reacting species, the enolate and methanol, are both nucleophilic, so to allow them to react a polarity inversion must take place. This was achieved by carrying out the reaction in the presence of 1-fluoro-3,3-dimethyl-1,3-dihydro-1$\lambda^3$-benzo[d][1,2]iodaoxole. By this approach, α-methoxy ketones were formed in high yields as single constitutional isomers under very mild conditions. Importantly, and in contrast with other methods that require the use of electrophilic sources of oxygen, a simple alcohol species can be used as a nucleophile in this reaction. The reaction also works with the carbonyl oxygen of ketones or amides acting as nucleophiles in an intramolecular fashion, yielding highly functionalized 3(2H)-furanones. Thus, this approach offers a route to these heterocyclic compounds, which are important structural units in medicinal chemistry, starting from simple and readily available substrates. The mechanisms for the inter- and intramolecular reactions were studied computationally. We found that the C$_\alpha$-OMe products were formed through a reductive elimination reaction (or ligand coupling) between the C-bound I(III)-enolate and methanol, and that the TFE additive activates the substrate through hydrogen bonding promoting the C−O bond formation. We also found that the intramolecular process that leads to the formation of furanones follows a similar mechanism.

## Methods

**General procedure for the catalytic reaction.** The corresponding allylic alcohol (0.3 mmol, 1 equiv.) was dissolved in a mixture MeOH/TFE (3:1) (15 mL). KBF$_4$ (30 mg, 0.24 mmol, 80 mol%), 1-fluoro-3,3-dimethyl-1,3-dihydro-1-λ-3-benzo[d][1,2]iodaoxole (102 mg, 0.36 mmol, 1.2 equiv.), and [Cp*IrCl$_2$]$_2$ (6 mg, 0.0075 mmol, 0.025 equiv.) were added and the mixture stirred at 35 °C during 2 h. After that, H$_2$O was added to dilute the reaction and the mixture was extracted with Et$_2$O (3 × 5 mL). The combined organic phases were dried with MgSO$_4$ and the solvent was evaporated under vacuum. The resulting crude was purified by column chromatography using petroleum ether/EtOAc (90:10) mixture as eluent.

**DFT calculations.** The calculations were carried out with the Gaussian 16 set of programs, using the M06 functional together with the 6-31G** basis sets for full structure optimization. IRC were done for both TS, where only one imaginary frequency was found and zero for the intermediate molecules. Then DEF2TZVPP in single point has been used with M06 for the energies. An implicit solvent model (IEFPCM, solvent = dimethyl formamide) was also used in all calculations (see the Supplementary Figs. 7 and 8, and Supplementary Data 1).

## Data availability

The authors declare that all data supporting the findings of this report are available in the article or in their supplementary information files. This includes experimental procedures, compound characterization and full computational details. Date are also available upon request from the corresponding author.

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

## Acknowledgements

This project was supported by the Swedish Research Council through Vetenskapsrådet and Formas, by the Knut and Alice Wallenberg Foundation (KAW 2016.0072), and by the Göran Gustafsson Foundation. A.S.-M. thanks Universitat de València, the Generalitat Valenciana, and the European Social Fund for a post-doctoral grant. We are also grateful to the European Funding Horizon 2020-MSCA (ITN-EJD CATMEC 14/06-721223). We also thank IZO-SGI SGIker of UPV/EHU for human and technical support. Open access funding provided by Stockholm University.

## Author contributions

B.M.-M. and A.S.-M. conceived and designed the project. B.M.-M. directed the project. A.S.-M. and S.M.-E. carried out experiments and prepared the Supporting Information. E.G.-B. and M.P. carried out all the mechanistic calculations and contributed to the mechanistic understanding. They also prepared the corresponding Supporting Information. All the authors discussed the results, and all participated in writing the article.

## Competing interests

The authors declare no competing interests.
