## [Peer Review File · Nature Communications]

Reviewers' comments:

Reviewer #1 (Remarks to the Author):

In this paper, the authors described a method to make alpha-methoxy ketone from allylic alcohols using a hypervalent iodine as an oxidant. Capitalizing on the Ir-catalyzed hydrogen transfer process, the authors generated enolates in a regioselective fashion, which is essential for the synthesis of the products in this paper. The authors did a reasonable condition optimization, performed thorough studies in scope expansion, and provided mechanistic insights with DFT calculations. The manuscript is well written. However, improvements should be made prior to publication in Nature Communications.

First, this work is an extension of the established methods reported by the same group. The authors described C-F bond and C-Cl bond forming reactions before. Here they report C-O bond formation. In order for this reaction to be useful, the scope of O-nucleophiles should be expanded to include more than methanol and ethanol.

Second, the authors have performed DFT calculations to understand the mechanism of this reaction. As well as the calculation is performed, I wonder the possibility of single-electron transfer process as a viable pathway for product formation. The authors may need to discuss this point in the paper.

Third, the SI is well written, but the NMR spectra are not in acceptable shape. The peaks of many ¹H NMR spectra should be taller.

Reviewer #2 (Remarks to the Author):

In the current manuscript, Martin-Matute and coll. reported a cascade reaction involving an isomerization reaction of allylic alcohol followed by the trapping of the in-situ generated enolate by a nucleophile. The synthesis of alpha-substituted carbonyl compounds via a reaction between an enolate and an electrophile is known. This group has already reported several preparations of alpha-functionalized ketones via the catalytic generation of enolates from allylic alcohols and the reaction of these intermediates with various electrophiles (including oxygen-based electrophilic species). The novelty in this work comes from the use of a nucleophilic species for the alpha-functionalization. This unprecedented strategy deserves to be published.

However, some explanations are missing or are not clear enough and have to be completed, more experiments have to be carried out as well, before accepting this publication.

1) In figure 2: why did the authors mention 0% conversion for several iodine(III) reagents when conversions were mentioned in the ESI? This statement provide misleading information. Did the

authors observed any radical addition with some iodide(III) species such as IV and V, which are known to be good CF₃ donors? Is the result obtained with VI and VII (see table in the ESI) surprising regarding their properties?

I have also to point out that yield was written in figure 2 instead of conversion. A yield corresponds to the quantity of isolated product, GC or NMR yield does not mean anything.

2) the alpha-fluoroketone 6 does not appear neither in the scheme nor in the table.

3) I do not understand the role of BF₄. Could the authors give more information on that?

4) KB₄ is known to promote anion metathesis, especially in polar protic solvent such as alcohols. So why [Cp*IrCl₂]₂ is an active pre-catalyst, when [Cp*Ir(H₂O)₃]₂SO₄ is not. The same species should be generated in both cases. Could the authors provide more explanation? The authors should also try to synthesize/characterize these species ([Cp*IrCl₂]₂ + KBF₄ and [Cp*Ir(H₂O)₃]₂SO₄ + KBF₄). Again, page 10, it was mentioned that KBF₄ speeds up the reaction, but no explanation/conclusion for this observation was given.

5) page 7, line 6: the sentence is missing after “, and”.

6) Why is the nucleophile limited to methanol? Is it related to the nucleophilicity of methanol compared to the other alcohols? The authors have to comment on that and also provide more examples with various alcohols.

7) The general structure of the lactone in figure 4 is inaccurate.

8) page 9, line 11: compound 9g can not be isolated as a mixture of diastereomers

9) Figure 5, page 11: is the iridium complex playing any role in the alkylation? The authors should carry out an alkylation in the same conditions starting from a pre-formed enolate and without iridium complex. The enolate is a strong base; so how can the authors propose the formation of fluorhydric acid without any protonation of the enolate (this protonation can also be done by methanol or TFE)? Some calculations have been realized, but what is the rate limiting step? The formation of B or the “reductive elimination”? I suggest to the authors also to calculate the same pathway with at least one of non-reactive iodine(III) to show the difference in energy in the key steps.

Reviewer #3 (Remarks to the Author):

Recommendation: Publish in Nature Communication after major revisions.

Comments:

The paper reports an unprecedented selective umpolung strategy for the synthesis of carbonyl compounds. The authors show that this nucleophile, formed in a catalytic fashion, is able to react with other nucleophiles, such as methanol, in a process mediated by an iodine(III) reagent. More importantly, they carried out both experimental and computational investigations, and mechanisms are proposed for both the inter- and intramolecular reactions, explaining the key role of the iodine(III) reagent in this umpolung approach. The manuscript is written well and the work

is competently executed. Therefore, I recommend its publication in Nature Communication as an article.

Further major points:

1. Some errors are shown in the article, such as ", and." in line 6 on page 7. And English should be improved.

2. The basis set shown in the manuscript does not match with the one in the supporting information, it needs to be further checked for correctness.

3. The word "rejected" should be replaced by "excluded" in line 10 of the 3rd paragraph on page 10.

4. How does KBF₄ accelerate the conversion of 1 to species A, the detailed mechanism should be provided.

5. In Figure 5, why the catalyst [Cp*IrCl₂]₂ only works in 1,3-H shift process and in the rest steps only I(III) plays a key role, how to prove the action of these two catalysts occurs in sequence or simultaneous in the title reaction. These two mechanisms should be compared.

6. And the paper shows that a molecule TFE can lower the activation barrier of TS1, how about two or three TFE molecules?

7. The format of the transition state TS3 shown in Figure 6 should be revised as to TS2.

Reviewers' comments:

Reviewer #1 (Remarks to the Author):

In this paper, the authors described a method to make alpha-methoxy ketone from allylic alcohols using a hypervalent iodine as an oxidant. Capitalizing on the Ir-catalyzed hydrogen transfer process, the authors generated enolates in a regioselective fashion, which is essential for the synthesis of the products in this paper. The authors did a reasonable condition optimization, performed thorough studies in scope expansion, and provided mechanistic insights with DFT calculations. The manuscript is well written. However, improvements should be made prior to publication in Nature Communications.

1.1 First, this work is an extension of the established methods reported by the same group. The authors described C-F bond and C-Cl bond forming reactions before. Here they report C-O bond formation. In order for this reaction to be useful, the scope of O-nucleophiles should be expanded to include more than methanol and ethanol.

Answer: During the last years, in our group we have reported the selective functionalization of iridium enolates with several electrophilic halogenated reagents. This methodology was expanded last year by the use of electrophilic oxygenated agents to achieve unsymmetrical aliphatic acyloins (reference 40).

Herein, we have described a new concept. This is an unprecedented umpolung strategy, where the iridium enolates (nucleophiles) react with nucleophiles, instead of electrophiles. This is the novelty of the reaction, and we have shown this with oxygen nucleophiles, in the form of methanol and ethanol (Figure 3) or in the form of a carbonyl oxygen (Figure 4). Nevertheless, we have now also included propanol obtaining the desired product (substrate **7a**).

1.2 Second, the authors have performed DFT calculations to understand the mechanism of this reaction. As well as the calculation is performed, I wonder the possibility of single-electron transfer process as a viable pathway for product formation. The authors may need to discuss this point in the paper.

Answer: Several radical scavengers have been tested in our reaction conditions. The reaction proceeds well in presence of TEMPO and diphenylethylene as additives, which may indicate that the reaction does not follow a radical pathway. A sentence in the manuscript has been included and a new scheme has been added to the Supplementary document (Supplementary Figure 6).

1.3 Third, the SI is well written, but the NMR spectra are not in acceptable shape. The peaks of many ¹H NMR spectra should be taller.

Answer: The ¹H NMR spectra are now presented as suggested by the referee, and we have also zoomed in some regions of the spectra.

Reviewer #2 (Remarks to the Author):

In the current manuscript, Martin-Matute and coll. reported a cascade reaction involving an isomerization reaction of allylic alcohol followed by the trapping of the in-situ generated enolate by a nucleophile. The synthesis of alpha-substituted carbonyl compounds via a reaction between an enolate and an electrophile is known. This group has already reported several preparations of alpha-functionalized ketones via the catalytic generation of enolates from allylic alcohols and the reaction of these intermediates with various electrophiles (including oxygen-based electrophilic species). The novelty in this work comes from the use of a nucleophilic species for the alpha-functionalization. This unprecedented strategy deserves to be published.

However, some explanations are missing or are not clear enough and have to be completed, more experiments have to be carried out as well, before accepting this publication.

2.1 In figure 2: why did the authors mention 0% conversion for several iodine(III) reagents when conversions were mentioned in the ESI? This statement provides misleading information.

Answer: The yields mentioned in figure 2 refer only to the desired compound **2a**. We have now clarified in Figure 2 that the reported yields refer to **2a**. In the figure footnote, we also refer to the supplementary document for further details.

2.2 Did the authors observed any radical addition with some iodide(III) species such as IV and V, which are known to be good CF₃ donors?

Answer: As the referee points out, iodine species **IV** and **V** are well known to be able to donate CF₃ groups via radical pathways. In our reaction conditions, these reagents did not yield any radical addition. In fact, only decomposition of the allylic alcohol was observed.

2.3 Is the result obtained with VI and VII (see table in the ESI) surprising regarding their properties?

Answer: Under these conditions, compound **VII** promoted decomposition of allylic alcohols **1a** to a large extent. This is not surprising, due to the oxidative character of **VII**. On the other hand, with reagent **VI** decomposition was not observed.

2.4 I have also to point out that yield was written in figure 2 instead of conversion. A yield corresponds to the quantity of isolated product, GC or NMR yield does not mean anything.

Answer: We agree with the referee, and yields measured by NMR spectroscopy might be meaningless if one calculates them just by for comparison with remaining starting material. One might, in this case, miss that the starting material maybe decomposes. However, we have measured the yields against an internal standard (1,2,4,5-tetrachloro-3-nitrobenzene). We also show that this standard method is reliable in our case, since the isolated yields are comparable with those measured by NMR against the internal standard (Figures 3 and 4).

2.5 2) the alpha-fluoroketone 6 does not appear neither in the scheme nor in the table.

Answer: alpha-Fluoroketone **6** was only observed under slightly different reaction conditions, but never under the reaction conditions shown in Table 1. We thank the referee for this observation, as it might be confusing to show it. We have now removed it.

2.6 3) I do not understand the role of KBF₄. Could the authors give more information on that?

Answer: We cannot assign a clear and unique role to KBF₄. We indeed also questioned this, and tested the isomerization of allylic alcohol **1a** with and without KBF₄, and this was presented in our first submission (Supplementary Figure 1). A faster reaction rate was measured with KBF₄ than in its absence. Besides, we have now also tested the reaction with a pre-formed silyl-enolate with and without the additive and observed no difference (Supplementary Figure 2). Taking all these results into account, we believe that the additive only has a role in accelerating the formation of the active iridium enolate species (**A**).

2.7 4) KBF₄ is known to promote anion metathesis, especially in polar protic solvent such as alcohols. So why [Cp*IrCl₂]₂ is an active pre-catalyst, when [Cp*Ir(H₂O)₃SO₄] is not. The same species should be generated in both cases. Could the authors provide more explanation?

Answer: Previous investigations by our group using EXAFS with these Ir^{III} complexes showed that a halide ligand within the coordination sphere of the metal is needed to catalyze the 1,3-H shift (ref 45). Similar investigations on hydrogen transfer processes by Nguyen et. al. (*J. Am. Chem. Soc.* **2015**, *137*, 4151-4157) using iridium complexes also points to the need of the halide ligand. Complex [Cp*Ir(H₂O)₃SO₄] lacks the halide ligand in the structure, and it is not active in the isomerization of allylic alcohols (reference 45). However, when the halide ligand is provided, by a halogenating reagent (*N*-chlorosuccinimide), the reaction yields the product. This has been added to the manuscript (page 5), and to the SI (Supplementary Figure 4).

2.8 The authors should also try to synthesize/characterize these species ([Cp*IrCl₂]₂ + KBF₄ and [Cp*Ir(H₂O)₃SO₄] + KBF₄). Again, page 10, it was mentioned that KBF₄ speeds up the reaction, but no explanation/conclusion for this observation was given.

Answer: See answer to comment 2.6 and 2.7. We have however noted this comment, and we will attempt to characterize these intermediates provided we obtain beam time to perform XAS experiments, as important valuable information may be found. However, we believe they would not change significantly the scientific advances presented in this communication.

2.9 5) page 7, line 6: the sentence is missing after “, and”.

Answer: It was meant to finish the sentence before “, and”. We have now removed it.

2.10 6) Why is the nucleophile limited to methanol? Is it related to the nucleophilicity of methanol compared to the other alcohols? The authors have to comment on that and also provide more examples with various alcohols.

Answer: We showed that ethanol is also able to act as a nucleophile in our first submission (40% yield, Figure 3). Further, the oxygen of a carbonyl group also acts as a nucleophile (see Fig. 4, 11 examples). We also show now propanol as a new entry (Figure 3, 20%). Therefore, other nucleophiles can be used, although further optimization would be needed. We intend to continue these investigations and report our findings in due course.

2.11 7) The general structure of the lactone in figure 4 is inaccurate.

Answer: We thank the referee for noticing this error. The structure of the lactone has been modified.

2.12 8) page 9, line 11: compound 9g can not be isolated as a mixture of diastereomers

Answer: Indeed, we thank the referee again for this important observation. **9g** is not a mixture of diastereoisomers. We have removed the typo “both”.

2.13 9) Figure 5, page 11: is the iridium complex playing any role in the alkylation? The authors should carry out an alkylation in the same conditions starting from a pre-formed enolate and without iridium complex.

Answer: The reaction has been carried out using phenyl silyl enol ether as starting material in the absence of the iridium catalyst. The desired product was obtained in quantitative yield proving that the iridium complex was not involved in the alkylation part (these results have been included in the SI, Supplementary Figure 3).

2.14 The enolate is a strong base; so how can the authors propose the formation of fluorhydric acid without any protonation of the enolate (this protonation can also be done by methanol or TFE)?

Answer: Our intention was to show that formally 1 equiv. of F⁻ and 1 equiv. of H⁺ is formed. In a polar solvent these species would be stabilized by an extensive hydrogen bonding network. We thank the referee for this comment, as the reaction scheme was misleading.

2.15 Some calculations have been realized, but what is the rate limiting step? The formation of B or the “reductive elimination”? I suggest to the authors also to calculate the same pathway with at least one of non-reactive iodine(III) to show the difference in energy in the key steps.

Answer: According to our deuterium labeling investigations (Supplementary Figure 5), the 1,3-H shift may not be involved in the step that is rate limiting. This fact would suggest that the rls is a step involving the I(III) species. This has now been added to the manuscript. In addition, we are currently writing an article where we present a very extensive study combining theoretical and experimental investigations dealing with these iridium-catalyzed isomerizations. Nevertheless, the results agree with the rls of the reactions coming after the 1,3-hydride shift.

Reviewer #3 (Remarks to the Author):

Recommendation: Publish in Nature Communication after major revisions.

Comments:

The paper reports an unprecedented selective umpolung strategy for the synthesis of carbonyl compounds. The authors show that this nucleophile, formed in a catalytic fashion, is able to react with other nucleophiles, such as methanol, in a process mediated by an iodine(III) reagent. More importantly, they carried out both experimental and computational investigations, and mechanisms are proposed for both the inter- and intramolecular reactions, explaining the key role of the iodine(III) reagent in this umpolung approach. The manuscript is written well and the work is competently executed. Therefore, I recommend its publication in Nature Communication as an article.

Further major points:

3.1 1. Some errors are shown in the article, such as “, and.” in line 6 on page 7. And English should be improved.

Answer: “, and” has been removed.

3.2 2. The basis set shown in the manuscript does not match with the one in the supporting information, it needs to be further checked for correctness.

Answer: We thank the referee for this observation. This has now been corrected.

3.3 3. The word “rejected” should be replaced by “excluded” in line 10 of the 3rd paragraph on page 10.

Answer: The word “rejected” has been replaced by “excluded”.

3.4 4. How does KBF₄ accelerate the conversion of 1 to species A, the detailed mechanism should be provided.

Answer: See answer 2.6 above.

3.5 5. In Figure 5, why the catalyst [Cp*IrCl₂]₂ only works in 1,3-H shift process and in the rest steps only I(III) plays a key role, how to prove the action of these two catalysts occurs in sequence or simultaneous in the title reaction. These two mechanisms should be compared.

Answer: we have tested the reaction with a pre-formed enolate with and without the Iridium catalyst and observed no improvement in the reaction (Supplementary Figure 3). Thus, we believe that the iridium catalyst only has a role in in 1,3-H shift process for the formation of the active iridium enolate species (**A**).

3.6 6. And the paper shows that a molecule TFE can lower the activation barrier of TS1, how about two or three TFE molecules?

Answer: The introduction of a single molecule of TFE stabilizes the most densely charged negative atoms of the transition state, decreasing the energy, and serves as a proof of the positive effect of TFE in the reaction. TS1 has been calculated with two molecules of TFE (Supplementary Figure 8). The energy of TS1 with two molecules of TFE is 20 kcal/mol instead of 16 kcal/mol with one molecule of TFE. The addition of more solvent molecules usually has no further evident positive effect (computationally) for two reasons. The stabilization is not so significant with the second/third molecules, and is easily counterbalanced by the negative entropic effect of adding multiple molecules in a single structure. Also, adding more than one solvent molecules introduce the computational problem of increasing the number of possible structures to compute, through an strong increase in the number of positions and conformations attainable by the combination of molecules.

3.7 7. The format of the transition state TS3 shown in Figure 6 should be revised as to TS2.

Answer: The format of the transition state TS3 have been modified (Figure 6).

Editorials' comments:

The following changes have not been highlighted in the manuscript

Title does not contain punctuation: The title has been changed. To eliminate the colon, the new title is: **"An Unprecedented Umpolung Strategy to React Catalytic Enols with Nucleophiles"**

Abstract: The abstract has been rewritten and it is in accordance with the guidelines.

Main text: The Method and Data availability sections have been included. Additionally, the sections are following the required order.

Figures: The chemical structures have been changed according to the Style guide (avoiding red and green). Moreover, Figure 1 shows now different panels which are labelled with a single letter.

Results: Format of references to supplementary items have been corrected.

Supplementary information: The supplementary items have been labelled correctly and the references have been moved to the end of the document.

REVIEWERS' COMMENTS:

Reviewer #1 (Remarks to the Author):

The authors made some revision to this manuscript. However, they failed to expand the scope of alcohol substrates, even though this point was raised by two reviewers. This limitation significantly diminishes the utility of this method.

This reviewer is not convinced of the novelty of this current methodology either, since oxidative coupling is a broadly utilized strategy in synthesis in recent years.

With that, this reviewer thinks this manuscript may fall short of the requirement of Nature Communications.

Reviewer #2 (Remarks to the Author):

the authors took into account all the comments and requests for corrections made by the referees. They provided reasonable answers and consequently I recommend the publication of this work in Nature Communication

Reviewer #3 (Remarks to the Author):

The paper reports an unprecedented selective umpolung strategy for the synthesis of carbonyl compounds. The authors show that this nucleophile, formed in a catalytic fashion, is able to react with other nucleophiles, such as methanol, in a process mediated by an iodine(III) reagent. More importantly, they carried out both experimental and computational investigations, and mechanisms are proposed for both the inter- and intramolecular reactions, explaining the key role of the iodine(III) reagent in this umpolung approach. The manuscript is written well and the work is competently executed. And they answered carefully the questions proposed by the reviewers. Now it can be accepted and published in Nature Communications.